# Constructing Machine-Precision Neural Networks with Quasi-Interpolants

## Abstract

Neural networks struggle to train to machine precision for simple interpolation tasks, limiting their use in scientific computing pipelines. We address this by providing the first explicit MLP construction that provably achieves machine-precision interpolation with $\log(1/\varepsilon)$ parameter scaling — matching classical polynomial methods — while remaining implementable in floating-point arithmetic. Our construction, based on quasi-interpolation theory, reveals a critical bandwidth parameter $\lambda$ that controls a tradeoff between aliasing error and conditioning; setting the optimal $\lambda$ implies weight magnitudes must grow with width. Using this framework to analyze trained MLPs, we find that they do not maintain the required weight scaling and exhibit rank saturation. Our results provide a principled framework for understanding why optimization, not expressivity, underlies precision failures in scientific machine learning.

## 1 Introduction

Machine learning has the potential to dramatically accelerate scientific workflows by replacing expensive physics-based subroutines with learned surrogates, from neural operators for weather modeling to surrogates for aerodynamics simulations (Pathak et al., 2022; Li et al., 2025; Alkin et al., 2025; Mao et al., 2024). However, numerical precision remains a central barrier to deployment: many scientific pipelines require tight residual tolerances and stable long-horizon rollouts, yet current methods struggle to reliably reach the required fidelity (McGreivy & Hakim, 2024). In fact, neural networks already struggle to reach machine precision in the simple controlled setting of noise-free interpolation of smooth functions from exact samples (Michaud et al., 2023a; Liu et al., 2025a). Even for analytic 1D targets like $f(x) = \sin(2\pi x)$, standard training stalls 16-22 *orders of magnitude* above `fp64` limits ($10^{-32}$), plateauing around $10^{-6}$ to $10^{-10}$ relative error (Figure 1b).

Surprisingly, increasing model size or training time does not reliably yield additional digits of accuracy (Wang et al., 2024; Liu et al., 2025b). Current techniques do not characterize how precision can successfully *scale* with increased model capacity or compute. This stands in contrast to numerical analysis, where decades of work provide provable convergence guarantees (Trefethen & Bau, 2022; Boyd, 1989), and to modern foundation models, where empirical scaling laws predictively relate performance to model size (Hoffmann et al., 2022; Kumar et al., 2024). Understanding whether precision can scale predictably with capacity is therefore a foundational question to unlock broader adoption of machine learning across scientific domains.

Motivated by recent diagnostic work (Michaud et al., 2023a; Wang & Lai, 2023), we study multi-layer perceptrons (MLPs) in the noise-free 1D interpolation regime to understand why precision fails to scale with model size and compute. We decompose this problem into two questions:

1. **Expressivity**. What machine-precision representations are numerically realizable as MLPs? Universal approximation theorems (Hornik et al., 1989) guarantee that high-precision interpolating MLPs exist, but classical results either provide vacuous parameter bounds (achieving $\varepsilon$ error requires $\mathrm{poly}(1/\varepsilon)$ parameters) or require constructions with exponentially large weight magnitudes (Mhaskar, 1993). We lack explicit representations that are simultaneously (i) machine-precise, (ii) parameter-efficient ($\log(1/\varepsilon)$ scaling), and (iii) numerically realizable in finite-precision arithmetic.

2. **Optimization**. Why does training fail to discover high-precision solutions? Even when numerically stable representations exist in principle, training often fails to find them. We lack understanding of

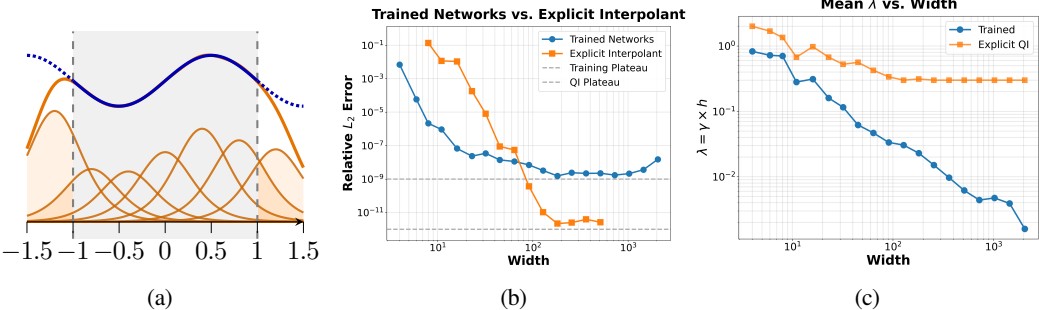

(a)                   (b)                   (c)

Figure 1: **Left (a):** Quasi-interpolants represent $f$ as a sum of localized, translated kernels, with additional "halo" nodes outside $[-1,1]$ reduce boundary error (shaded). **Middle (b):** Relative $\ell_2$ error versus width: the explicit QI interpolant continues improving to the fp64 floor, while end-to-end trained MLPs plateau several orders of magnitude higher. **Right (c):** Dimensionless bandwidth $\lambda = \gamma h$ versus width: the QI construction maintains roughly constant $\lambda$, whereas trained networks drive $\lambda \to 0$ with width, indicating a length-scale mismatch relative to the construction.

*why* trained networks converge to sub-optimal solutions, and *how* attainable precision depends on the interplay between width, optimizer choice, and training dynamics.

We address both gaps by constructing explicit MLP interpolants using tools from quasi-interpolation theory (Trefethen, 2019; Buhmann & Jäger, 2022). Our contributions are:

- **We provide an explicit construction for high-precision MLP interpolation.** Ours is the first MLP construction that provably achieves machine-precision interpolation with $\log(1/\varepsilon)$ parameter scaling, matching classical polynomial methods. Unlike previous universal approximation results, our construction remains implementable in floating-point arithmetic, yielding a realizable route to machine precision with provable scaling. Critically, our construction also exposes a *bandwidth parameter $\lambda$* that must scale with network width to balance aliasing error against conditioning. This shows that weight magnitudes must grow with the number of parameters to maintain numerical stability at high precision.

- **We study why optimization fails to discover high-precision solutions.** Surprisingly, trained MLPs with standard second-order optimizers (Urbán et al., 2025) achieve geometric convergence at small widths, but plateau as capacity increases – the opposite of expected scaling behavior. Our quasi-interpolant construction provides a diagnostic lens: we find that (i) trained MLPs learn sparse, non-uniform representations that violate the conditioning requirements exposed by our construction, and (ii) learned length-scales collapse rather than growing with width.

These results represent a first step towards a principled framework for understanding and achieving machine-precision interpolation with neural networks.

## 2  RELATED WORK

**High-precision scientific ML**    Precision ceilings in `fp64` training are increasingly recognized as a bottleneck in scientific ML. Across settings ranging from function regression to operator learning, researchers report plateaus far above floating-point limits that persist despite increased model capacity, compute, or data, motivating a range of responses (Michaud et al., 2023b; Wang & Lai, 2023; Cao et al., 2023). Prior attempts to improve the precision floor broadly span architectural changes, optimizer design, and refinement-based training paradigms:

- **Architecture.** Architectural modifications raise attainable precision by mitigating spectral bias (improving high-frequency fitting) or reshaping training geometry to improve conditioning and capacity allocation. Recent works attempt to address spectral bias via sinusoidal activations with principled initialization (Sitzmann et al., 2020), reweight objectives to focus capacity on high-error regions using residual-adaptive mechanisms (Wang et al., 2024), and expose precision-conditioning tradeoffs through barycentric-weight parameterizations (Liu et al., 2025a).

- **Optimizer.** Second-order and geometry-aware optimizers improve precision by providing better-conditioned update directions on ill-conditioned loss landscapes. Recent works attempt to stabilize Shampoo-style preconditioning by operating in the preconditioner's eigenbasis (Vyas et al., 2025; Gupta et al., 2018), leverage natural-gradient structure for faster convergence (Müller & Zeinhofer, 2023), and apply quasi-Newton refinement to push PINN training toward lower residuals (Urbán et al., 2025; Kiyani et al., 2025).

- **Training paradigms.** Other works propose to decompose or reframe the learning problem into better-behaved alternatives. For example, Wang & Lai (2023) attempt to train successive networks on spectrum-normalized residuals (Wang & Lai, 2023), and Bacho et al. (2025) learn surrogates for Cholesky factors of Gauss–Newton operators to amortize solver structure.

Although these advances show that higher precision is achievable, they do not yet explain (i) how attainable precision should scale with width/compute for a single end-to-end MLP in the interpolation setting; and (ii) the training pathologies that yield diminishing or non-monotone returns even when more accurate solutions exist. We address both gaps using a classical constructive approximation framework, quasi-interpolation, that makes the relevant length scales and conditioning constraints explicit and links them to the optimization dynamics seen in practice.

**Interpolation and quasi-interpolation**   Classical polynomial interpolation achieves exponential convergence for analytic targets—a rate that is minimax optimal (Trefethen, 2019)—but becomes ill-conditioned as degree increases. Quasi-interpolation sidesteps this by replacing global polynomials with weighted sums of localized kernels, providing explicit stability guarantees through cardinal function constructions (Buhmann & Jäger, 2022). We show that one-hidden-layer MLPs with smooth activations (e.g., tanh, GELU) can exactly represent such quasi-interpolants, with the activation's derivative determining the underlying kernel.

**Universal approximation theorems for MLPs**   Universal approximation theorems guarantee that one-hidden-layer MLPs can approximate continuous functions arbitrarily well (Cybenko, 1989). Classical constructions require width scaling as $O(1/\varepsilon)$ or worse, which is infeasible in practice – this implies requiring $\gtrsim 10^{16}$ parameters for `fp64` machine precision. For analytic targets, Mhaskar (1993) and De Ryck et al. (2021) establish MLP constructions whose precision convergence exponentially with width. However, these constructions involve numerically computing $W$-th order derivatives for $W$ neurons. We show in Table 1 that these constructions demand weight magnitudes that far exceed the range of `fp64`.

## 3   CONSTRUCTING HIGH-PRECISION MLPS WITH QUASI-INTERPOLANTS

In this section, we describe our quasi-interpolation based construction for high-precision MLP interpolants. Our main theoretical result is Theorem 1, an explicit error decomposition with geometric parameter convergence. Crucially, our construction is the first to achieve these rates while remaining implementable in floating-point arithmetic (Table 1). We show that one-hidden-layer MLPs with standard activations can implement our constructions. Our analysis identifies a critical parameter $\lambda$, which controls a fundamental tradeoff between aliasing error and numerical conditioning. We show that choosing an optimal $\lambda$ requires the magnitudes of our construction's weights to grow with the number of neurons, contrary to typical initialization schemes.

We proceed by defining the quasi-interpolant and its finite-interval truncations (Section 3.1), showing its realization as a one-hidden-layer MLP (Section 3.2), and finally deriving an error decomposition that makes the geometric convergence rate explicit (Section 3.3).

### 3.1   CONSTRUCTING FOURIER-NORMALIZED QUASI-INTERPOLANTS

We begin by constructing an operator that interpolates a smooth function from uniform samples while maintaining stable conditioning as resolution increases. Quasi-interpolation provides an explicit, translation-invariant form: it reconstructs $f$ as a weighted sum of localized kernels centered at the sample locations. Our goal is to define the cardinal function $L_h$ in a way that (i) yields predictable convergence as the grid refines, and (ii) remains numerically implementable in finite precision on a bounded domain.

**Overview.** Let $\Omega = [-1,1]$ and let $f$ be known through uniform samples on a grid of spacing $h$. On the infinite lattice, a translation-invariant quasi-interpolant has the form:

$$(Q_h f)(x) := \sum_{k \in \mathbb{Z}} f(x_k) L_h(x - x_k), \qquad x_k := -1 + kh, \tag{1}$$

where $L_h$ is a cardinal function constructed by Fourier normalization of a kernel family. On the finite interval $\Omega$ we implement a *finite* operator by introducing: (i) a *halo* size $R$ limiting the number of centers placed outside $\Omega$, and (ii) a *Fourier stencil* half-width $K_c$ from truncating Fourier coefficients in the cardinal function construction. The behavior of our approximant is governed by the dimensionless bandwidth parameter

$$\lambda := \gamma h, \tag{2}$$

which couples the kernel bandwidth $\gamma$ with the grid scale $h$. This dimensionless parameter controls a fundamental tradeoff between localization and aliasing, made precise in Theorem 1.

**Grid and kernel.** Fix $N \in \mathbb{N}$ and define the step size and nodes

$$h := \frac{2}{N}, \qquad x_k := -1 + kh, \qquad k \in \mathbb{Z}. \tag{3}$$

For a halo size $R \in \mathbb{N}$, define the extended index range $I_{R,K_c} = \{-R - K_c, ..., N + R + K_c\}$ and the extended interval $\Omega_R := [-1 - Rh, 1 + Rh]$. Let $K : \mathbb{R} \to \mathbb{R}$ be a base kernel and define its scaled version

$$K_\gamma(x) := \gamma K(\gamma x), \qquad \gamma > 0, \qquad \lambda := \gamma h. \tag{4}$$

We denote the Fourier transform of $g$ by $\widehat{g}$.

**Constructing $L_h$ by Fourier normalization.** Define the Fourier character:

$$\widehat{C}_h(\omega) := \frac{h}{D_h(\omega)} = \frac{h}{\sum_{m \in \mathbb{Z}} \widehat{K}_\gamma\left(\omega + \frac{2\pi m}{h}\right)} = \sum_{j \in \mathbb{Z}} c_j e^{-ijh\omega}, \tag{5}$$

where $(c_j)_{j \in \mathbb{Z}}$ denote the Fourier coefficients and we assume the denominator $D_h$ is non-zero on $\mathbb{R}$. We define the stencil truncated cardinal function:

$$L_h^{(K_c)}(x) := \sum_{|j| \leq K_c} c_j K_\gamma(x - jh), \tag{6}$$

which approaches $L_h$ as $K_c \to \infty$.

**Finite operator on $\Omega$.** Consider samples $f(x_k)$ on the extended index set $I_{R,K_c}$. The halo- and stencil-truncated quasi-interpolant is[1]

$$(Q_{h,R}^{(K_c)} f)(x) := \sum_{k=-R}^{N+R} f(x_k) L_h^{(K_c)}(x - x_k), \qquad x \in \Omega. \tag{7}$$

Re-indexing yields the equivalent single-kernel sum

$$(Q_{h,R}^{(K_c)} f)(x) = \sum_{m=-R}^{N+R} a[m] K_\gamma(x - x_m), \qquad a[m] := \sum_{|j| \leq K_c} c_j f(x_{m-j}), \tag{8}$$

which defines a kernel network in the quasi-interpolant framework.

---

[1]The extended index set does not require training samples outside the interpolation interval in MLP training. Although the quasi-interpolant utilizes these samples to stabilize boundary error, the MLPs can determine arbitrary function values on the extended set to correct boundary error.

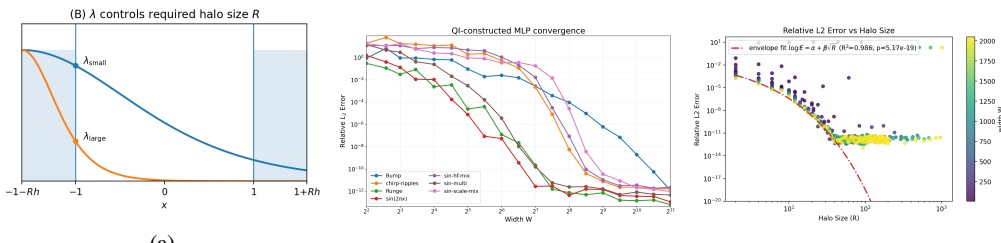

(a)

Figure 2: **Left (a):** $\lambda$ quantifies the bandwidth of the kernel $\gamma$ relative to the grid size $h$. $R$ halo nodes are appended to the boundary of the domain to avoid truncating boundary effects. For wider kernels (smaller $\lambda$), wider halo regions are needed for high accuracy. For larger $\lambda$, thinner halo regions can sufficiently account for the boundary effects. **Middle (b):** Convergence plots for QI-constructed MLP against different target functions. Regimes of geometric convergence for the difference target functions (Appendix A.2) are clearly visible, before the relative $L_2$ error plateaus near the conditioning boundary around $10^{-13}$. $\lambda$ is lower bounded by an algebraic blow-up, and we see the scaling depart the rate-theoretic rate. **Right (c):** Relative $L_2$ error for the same target across multiple sweeps. The error bounds predict a theoretical envelope, which a best-fit shown through the dashed red line. The error eventually plateaus around the conditioning floor of the construction, close to numerical precision at larger widths.

### 3.2 MLPs CAN IMPLEMENT QUASI-INTERPOLANTS

Although certain MLP activation functions do not satisfy the kernel properties, often an order derivative of a smooth activation does (for example, $\mathrm{sech}^2 = \tanh'$). Assume there exists a smooth activation $\psi : \mathbb{R} \to \mathbb{R}$ and an order $r \in \mathbb{N}$ such that

$$K(u) = \psi^{(r)}(u). \tag{9}$$

Then, by the chain rule,

$$(Q_{h,R}^{(K_c)} f)(x) = \sum_{m=-R}^{N+R} a[m] K_\gamma(x - x_m) = \frac{\mathrm{d}^r}{\mathrm{d}x^r} \left( \sum_{m=-R}^{N+R} \frac{a[m]}{\gamma^{r-1}} \psi\big(\gamma(x - x_m)\big) \right). \tag{10}$$

Therefore, defining the one-hidden-layer MLP

$$g_{\mathrm{MLP}}(x) := \sum_{m=-R}^{N+R} w[m]\, \psi\big(\gamma(x - x_m)\big), \qquad w[m] := \frac{a[m]}{\gamma^{r-1}}, \tag{11}$$

we obtain the exact identity

$$g_{\mathrm{MLP}}^{(r)}(x) = (Q_{h,R}^{(K_c)} f)(x). \tag{12}$$

*Crucially, this shows that standard one-hidden-layer MLPs can exactly represent the quasi-interpolant construction—the expressivity is not the bottleneck.* In particular, if the quasi-interpolant is applied to sampled data of an $r$-th derivative, then $g_{\mathrm{MLP}}$ recovers the target up to an integration polynomial of degree at most $r - 1$:

$$\widetilde{f}(x) = p_{r-1}(x) + g_{\mathrm{MLP}}(x), \tag{13}$$

where $p_{r-1}$ can be fixed by boundary conditions. For $\tanh' = \mathrm{sech}^2$ this polynomial reduces to a simple bias. For GELU and Swish, it reduces to a linear layer.

### 3.3 ERROR ANALYSIS

We present a numerical error bound on our quasi-interpolant construction. All kernel regularity and non-degeneracy conditions, together with explicit prefactors, are deferred to Appendix A.1; these conditions hold for the kernels used in our experiments.

**Theorem 1** (Numerical error bound). *Let $\lambda = \gamma h$. Assume $f$ is analytic in a complex neighborhood of $\Omega$, and assume the Fourier normalizer $D_h$ is uniformly bounded away from zero on a fixed strip*

*(Appendix **??**). Then there exist constants $c_1, c_2, c_3, c_4, C_2 > 0$ and prefactors $C_1(\lambda), C_3(\lambda), C_4(\lambda) > 0$ such that, for all $x \in \Omega$,*

$$\|f - Q_{h,R}^{(K_c)} f\|_{L^\infty(\Omega)} \leq \underbrace{C_1(\lambda) e^{-\frac{c_1}{h}}}_{\text{resolution}} + \underbrace{C_2 e^{-\frac{c_2}{\lambda^p}}}_{\text{aliasing / normalization}} + \underbrace{C_3(\lambda) e^{-c_3 \lambda R}}_{\text{halo truncation}} + \underbrace{C_4(\lambda) e^{-c_4 \lambda K_c}}_{\text{fourier stencil truncation}} . \tag{14}$$

*The exponent $p$ depends on the kernel family: for $K = \mathrm{sech}^2$ one can take $p = 1$.*

Note that the prefactors diverge algebraically as $\lambda \to 0$, thus enforcing an effective lower bound.

Equation (14) separates four error mechanisms, each governed by a distinct scale:

- **Grid resolution** $h$: For analytic targets, the approximation error decays geometrically as $e^{-c_1/h}$. Since $h = 2/N$, this gives exponential convergence in the number of nodes $N$, which is the best-case scenario when all other errors are controlled.

- **Bandwidth ratio** $\lambda$: The aliasing/normalization term decays as $e^{-c_2/\lambda^p}$, *but* the prefactors $C_1(\lambda), C_3(\lambda), C_4(\lambda)$ blow up as $\lambda \to 0$. This enforces an effective lower bound: $\lambda$ cannot be made arbitrarily small without destroying the constants in the other terms. For machine precision, we require $\lambda \gtrsim (\log(1/\epsilon))^{1/p}$.

- **Halo thickness in kernel units** $\lambda R$: Boundary truncation error scales as $e^{-c_3 \lambda R}$. Critically, the relevant quantity is the *product* $\lambda R$, not $R$ alone. Given the lower bound on $\lambda$ from the previous bullet, achieving precision $\epsilon$ requires $R \gtrsim \frac{\log(1/\epsilon)}{c_3 \lambda}$—the viable range of $\lambda$ determines how many halo nodes are needed.

- **Stencil width in kernel units** ($\lambda K_c$): The cardinal function truncation error decays as $e^{-c_4 \lambda K_c}$. Again, the coupling through $\lambda$ means that $K_c$ must scale as $\gtrsim \frac{\log(1/\epsilon)}{c_4 \lambda}$ to maintain prefilter accuracy.

**The bandwidth tradeoff.**   Equation (14) exposes competing constraints. To achieve a target tolerance $\epsilon$:

1. The resolution term requires sufficient nodes. Since $h = 2/N$, making $C_1(\lambda) e^{-c_1/h} \lesssim \epsilon$ typically requires $N = \mathcal{O}(\log(1/\epsilon))$ (up to the $\lambda$-dependent prefactor).

2. The aliasing/normalization term favors smaller $\lambda$ (narrower kernels relative to the grid), since $e^{-c_2/\lambda^p}$ decays as $\lambda \downarrow 0$. However, the prefactors $C_1(\lambda), C_3(\lambda), C_4(\lambda)$ blow up as $\lambda \to 0$, reflecting instability in the Fourier normalization.

3. The halo and stencil terms depend on the *products* $\lambda R$ and $\lambda K_c$. For fixed $\lambda$, achieving $C_3(\lambda) e^{-c_3 \lambda R} \lesssim \epsilon$ and $C_4(\lambda) e^{-c_4 \lambda K_c} \lesssim \epsilon$ requires $R, K_c = \mathcal{O}(\log(1/\epsilon)/\lambda)$ (again up to prefactors).

Together, these constraints define a viable regime $\lambda \in [\lambda_{\min}, \lambda_{\max}]$ where normalization remains stable and truncation costs remain practical.

**The key scaling relationship:** As width $N$ increases (refining $h = 2/N \to 0$), maintaining $\lambda = \gamma h$ in the viable range $[\lambda_{\min}, \lambda_{\max}]$ requires the bandwidth parameter $\gamma$ to scale proportionally with $N$. In MLP terms (Equation (11)), this means both the feature scale $\gamma$ and the weight magnitudes $|w[m]| \sim \gamma^{1-r}$ must *increase* with the number of parameters—a scaling property that standard initialization schemes and gradient-based training systematically fail to preserve (Section 4).

**Comparison to prior constructions.**   Table 1 compares our construction to prior universal approximation results. Classical constructions (Cybenko, 1989) scale as $\Theta(\varepsilon^{-1})$ in parameters — for `fp64` machine precision, this implies $\gtrsim 10^{16}$ neurons. The construction of Mhaskar (1993) achieves $\widetilde{O}(\log(1/\varepsilon))$ parameters, matching polynomial rates, but computing $W$-th order finite differences for $W$ neurons yields weights scaling with $\varepsilon^{-1/k}$, which exceeds `fp64` dynamic range at high precision. In contrast, our construction achieves optimal parameter scaling with weight magnitudes growing only as $O(\log(1/\varepsilon))$. We show that our construction is numerically realizable in `fp64` in practice (Figure 2b).

**Numerical validation.**   We implement the construction and verify that, for an optimal $\lambda$ and modest $(R, K_c)$, the quasi-interpolant achieves `fp64`-level accuracy with predictable scaling in $N$ (Figure 2b). We then contrast this with end-to-end trained MLPs, which fail to maintain the required bandwidth and weight scaling, resulting in a precision plateau (Section 4).

| Method | #params vs. $\varepsilon$ | Bit complexity (working precision) | Weight magnitudes |
|---|---|---|---|
| Cybenko (1989) | $\Theta(\varepsilon^{-1})$ | $\Theta(\log(1/\varepsilon))$ | $\|W\|_{\max} = \Theta(\varepsilon^{-1}\log(1/\varepsilon))$ |
| Mhaskar (1993) | $\widetilde{O}\big(\log(1/\varepsilon)\log\log(1/\varepsilon)\big)$ | $\gtrsim \widetilde{\Theta}\big(\log(1/\varepsilon)\log\log(1/\varepsilon)\big)$ | $\|W\|_{\max} = \Theta(\varepsilon^{-1/k})$ (order $k$) |
| Quasi-interpolant (ours) | $O(\log(1/\varepsilon))$ | $O(\log(1/\varepsilon)+\log\log(1/\varepsilon))$ | $\|W\|_{\max} = O(\log(1/\varepsilon))$ |

Table 1: Comparison of parameter count, working-precision bit complexity, and weight magnitudes for prior MLP constructions (universal approximation theorems) on analytic targets.

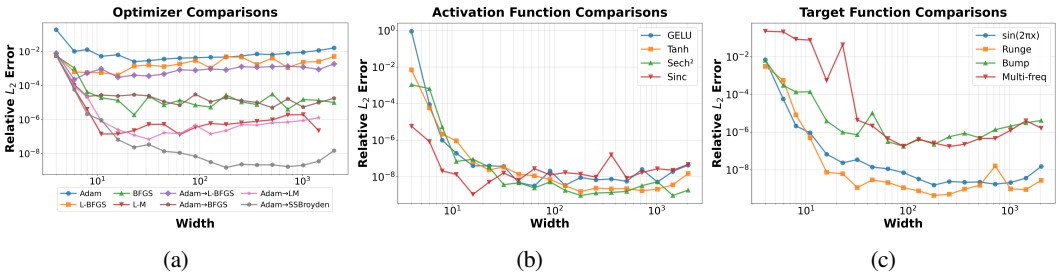

(a)           (b)           (c)

Figure 3: **Direct training plateaus.** Relative $\ell_2$ error versus width in `float64`. **Left (a):** Optimizer comparison on $f(x) = \sin(2\pi x)$ with $\mathrm{sech}^2$ activation ($W = 4$ to 2048). **Middle (b):** Activations under a fixed two-stage optimizer (Adam→SSBroyden) on the same target. **Right (c):** Width scaling across a range of target functions trained with $tanh$ activation and 2-stage optimization. Across settings, increasing width yields rapid initial gains followed by saturation well above machine precision.

## 4 EXPERIMENTS

Section 3 shows that machine-precision interpolation is achievable: the QI construction attains near round-off error with one-hidden-layer MLPs once the relevant length scales and normalization are set correctly. Yet in practice, direct training rarely reaches this regime (Figure 3).

We structure the remainder of this section around this gap. We (i) document the plateau phenomenon (Section 4.1), (ii) verify that the QI parameterization is `fp64`-realizable and achieves the predicted width–error scaling when the geometry is fixed (Section 4.2), and (iii) analyze how trained networks deviate from the construction (Section 4.3).

**Experimental setup.** We focus on noiseless 1D interpolation of smooth functions on $[-1,1]$, training on $n = 256$ uniformly spaced samples and evaluating relative $\ell_2$ error on a dense grid of 4096 points. All experiments use `fp64` precision.

### 4.1 DIRECT TRAINING FAILS TO REACH MACHINE PRECISION

Figure 3 reports relative $\ell_2$ error versus width for one-hidden-layer MLPs trained on noiseless samples from $f(x) = \sin(2\pi x)$. We quantify the precision ceiling across different optimizers, activations, and target functions. We observe that optimization plateaus across methods: in Figure 3(a), stronger optimizers reduce error by several orders of magnitude compared to Adam, but all tested configurations saturate far above `fp64` machine precision. Notably, widening beyond moderate width yields diminishing or even non-monotone returns. Additionally, plateaus persist across activations and targets: Figures 3(b–c) show the same qualitative behavior under different activations and target functions: rapid improvement at small widths followed by saturation well above machine precision.

### 4.2 MLPs CAN REALIZE QUASI-INTERPOLANTS

Before attributing the ceiling in Section 4.1 to expressivity, we verify that the QI construction is numerically realizable in `fp64` via gradient descent and exhibits the predicted width-to-error scaling when its geometric assumptions are enforced.

We implement the cardinal-kernel parameterization on a uniform grid as described in Equation (1) with a fixed grid and bandwidth $\gamma$, so training reduces to estimating only the output function evaluations of $f(x)$.

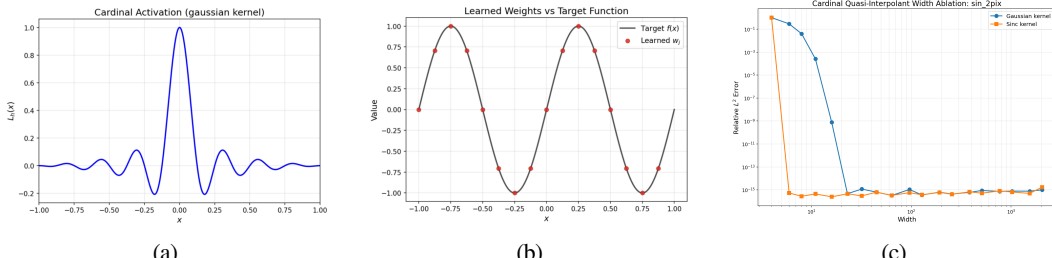

|     |     |     |
| --- | --- | --- |
| (a) | (b) | (c) |

Figure 4: **QI parameterization reaches `fp64` precision.** We fix the QI geometry and fit only the output coefficients in $f(x) \approx \sum_j w_j L_h((x - x_j)/h)$. **Left (a):** Localized cardinal basis function $L_h$. **Middle (b):** Fitted coefficients $w_j$ match the samples $f(x_j)$ at grid points. **Right (c):** Relative $\ell_2$ error versus width for Gaussian and sinc kernels, showing geometric convergence to the `fp64` floor.

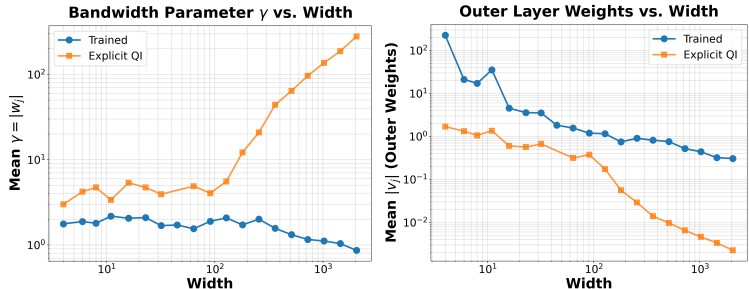

Figure 5: **Weight magnitude scaling** We compare the mean bandwidth $\gamma = |w_j|$ **(Left)** and mean absolute outer weights $|v_j|$ **(Right)** across network widths, where $v_j$ are the second-layer weights that combine activated features to produce $f(x)$. The blue curve shows trained networks obtained via Adam + SSBroyden optimization, and orange shows explicit quasi-interpolant (QI) constructions computed using theoretical formulas.

Figure 4 demonstrates that given a localized cardinal activation function $L_h(x)$ (panel a), (i) the learned coefficients $w_j$ closely match the target samples at grid points, confirming that the coefficient-recovery subproblem is numerically stable when the kernel geometry is specified correctly (panel b) and (ii) the resulting interpolant achieves geometric convergence to the `fp64` floor with increasing width (panel c). Thus, one-hidden-layer MLPs can reach machine precision under our QI construction.

Trained networks place centers outside the domain, qualitatively matching the required halo structure.

### 4.3 EMPIRICAL DIAGNOSIS

**Weight magnitude scaling.** The QI construction requires width-dependent parameter scaling to keep the dimensionless bandwidth $\lambda = \gamma h$ relatively constant as $h \sim 1/W$ decreases. Figure 5 shows a clear mismatch in magnitude scaling. The explicit QI increases the inverse length-scale $\gamma$ with width to keep the dimensionless bandwidth $\lambda = \gamma h$ roughly constant as $h \sim 1/W$ decreases. In contrast, end-to-end training keeps $\gamma = \mathcal{O}(1)$ (so $\lambda \to 0$) and uses much larger outer-layer weights than the QI solution across widths. We view this as evidence that training does not recover the QI normalization/geometry, consistent with a less stable (more ill-conditioned) interpolation regime.

This scaling mismatch suggests trained solutions rely more on cancellation among overlapping features, rather than the well-conditioned cardinal regime induced by the QI construction.

**Rank saturation.** MLPs often learn sparse representations (Frankle & Carbin, 2019), and we investigate if such sparse representations diverge from our construction.

We consider the induced linear system upon freezing the MLPs $\Phi a = b$, where $\Phi$ is the induced design matrix consisting of the $tanh$ activation functions evaluated over the training points, $a$ are the readout variables, and $b$ is the zero-mean target function evaluated at training points.

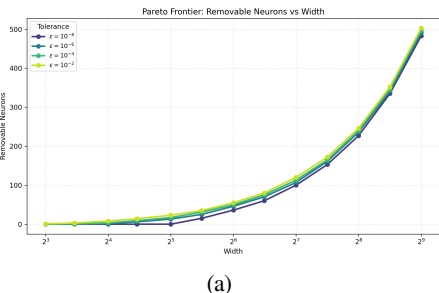
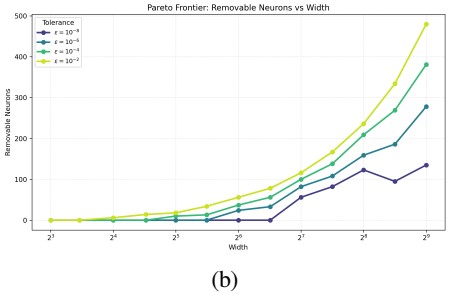

(a)                                          (b)

Figure 6: Pareto curves at different L2RE thresholds of removable neurons against the width for pruning the trained (**Right**) and the constructed (**Left**) MLPs using a greedy pruning algorithm. The constructed MLPs appear more sensitive to pruning, especially for sharper thresholds.

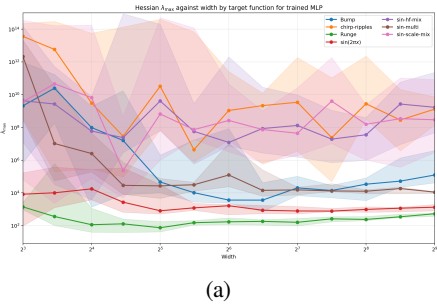
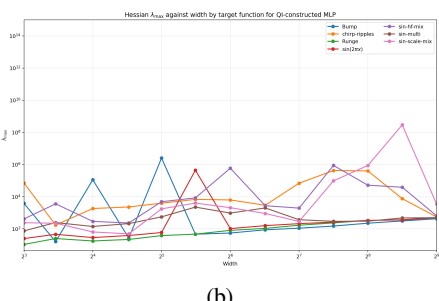

(a)                                          (b)

Figure 7: Hessian conditioning of the trained MLP (**Left**) and the QI-constructed MLP (**Right**). As seen, the constructed Hessians are better conditioned than the trained Hessians.

Figure 6 displays the pruning frontier of the trained and constructed MLPs at different relative $L_2$ error tolerances for the $sin(2\pi x)$ target. The pruning is determined by Orthogonal Matching Pursuit (OMP) (Pati et al., 1993), by greedily building subset columns of $\Phi$ to minimize the residual, and then re-computing the optimal least squares solution at each step. The QI-contructed MLPs appear more robust to sparsification than the trained MLPs, suggesting that sparsification and improper node utilization is holding back trained MLPs from reaching better precision. Furthermore, the trained MLPs display highly compressed frontiers, which suggests that a small subset of neurons are responsible for achieving most of the MLPs accuracy, and most extra neurons remain redundant, whereas the constructed MLP's nodes share a more even load in interpolation quality.

One may suggest that the reason the MLPs train these sparse representations is that they tend to yield better conditioned loss landscapes, and hence that conditioning gates the MLPs from matching the performance of the quasi-interpolants. We argue that this is not the case. Figure 7 shows the dominant eigenvalues of the trained and constructed MLPs. As seen, the constructed MLPs, although they display dominant growth, surprisingly remain better conditioned than the trained MLPs. Hence, conditioning is unable to account for the performance difference between trained and construed MLPs.

## 5 DISCUSSION

In this work, we introduce the first explicit MLP construction achieving machine-precision interpolation with $\log(1/\epsilon)$ parameter scaling while being realizable in floating-point arithmetic. This result demonstrates that optimization, not expressivity, is the bottleneck for high-precision neural networks. Our construction exposes a critical bandwidth parameter, $\lambda$, that governs an aliasing-conditioning tradeoff; analysis of trained MLPs suggests they fail to maintain the required scaling, exhibiting length-scale collapse and effective rank saturation. However, the conditioning of the constructed MLPs is not prohibitive to trained MLPs. Future work will develop optimization interventions, motivated by our theoretical insights, to enforce these representational constraints during training. We believe our quasi-interpolant framework represents a step towards principled scaling laws for high-precision scientific machine learning.

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

# A APPENDIX

## A.1 THEORETICAL RESULTS

### A.1.1 NOTATION

**Fourier transform.** For $f \in L^1(\mathbb{R})$, we use the convention

$$\widehat{f}(\omega) := \int_{\mathbb{R}} f(x)e^{-i\omega x}dx, \qquad f(x) = \frac{1}{2\pi}\int_{\mathbb{R}} \widehat{f}(\omega)e^{i\omega x}d\omega. \tag{15}$$

**Uniform grid, halo, and kernel scaling.** We work on $\Omega = [-1,1]$ with uniform step $h = 2/N$ and nodes $x_k = -1 + kh$. For $R \in \mathbb{N}$, define the halo interval $\Omega_R = [-1 - Rh, 1 + Rh]$ and halo indices $\{-R,...,N+R\}$. Let $K : \mathbb{R} \to \mathbb{R}$ be a kernel profile and define the scaled family

$$K_\gamma(x) := \gamma K(\gamma x), \qquad \gamma > 0, \qquad \lambda := \gamma h. \tag{16}$$

We use both continuous and sampled supremum norms:

$$\|f\|_{L^\infty(\Omega_R)} := \sup_{x \in \Omega_R} |f(x)|, \qquad \|f\|_{\ell^\infty(\Omega_R \cap h\mathbb{Z})} := \max_{-R \leq k \leq N+R} |f(x_k)|. \tag{17}$$

**Fourier-normalized character and quasi-interpolants.** Define the normalizer and Fourier character

$$D_h(\omega) := \sum_{m \in \mathbb{Z}} \widehat{K}_\gamma\left(\omega + \frac{2\pi m}{h}\right), \qquad \widehat{C}_h(\omega) := \frac{h}{D_h(\omega)}. \tag{18}$$

Let $\widetilde{C}_\lambda(\theta) := \widehat{C}_h(\theta/h)$ (a $2\pi$-periodic function of $\theta$), and let $(c_j)_{j \in \mathbb{Z}}$ denote its Fourier coefficients:

$$c_j := \frac{1}{2\pi}\int_{-\pi}^{\pi} \widetilde{C}_\lambda(\theta)e^{ij\theta}d\theta. \tag{19}$$

Define the quasi-cardinal and its stencil truncation

$$L_h(x) := \sum_{j \in \mathbb{Z}} c_j K_\gamma(x - jh), \qquad L_h^{(K_c)}(x) := \sum_{|j| \leq K_c} c_j K_\gamma(x - jh), \tag{20}$$

and the infinite, halo-, and halo+stencil-truncated operators (for $x \in \Omega$)

$$(Q_h f)(x) := \sum_{k \in \mathbb{Z}} f(x_k)L_h(x - x_k), \tag{21}$$

$$(Q_{h,R} f)(x) := \sum_{k=-R}^{N+R} f(x_k)L_h(x - x_k), \tag{22}$$

$$(Q_{h,R}^{(K_c)} f)(x) := \sum_{k=-R}^{N+R} f(x_k)L_h^{(K_c)}(x - x_k). \tag{23}$$

**Interior projector (Option A).** Fix $\varepsilon \in (0,1)$ and define the interior projector in Fourier space by

$$\widehat{(\mathcal{P}_{h,\varepsilon}f)}(\omega) := \mathbf{1}_{\{|\omega| \leq (1-\varepsilon)\pi/h\}}\widehat{f}(\omega). \tag{24}$$

### A.1.2 PROOF OF THEOREM 1

**Assumption 1** (Target regularity). *There exist $\rho > 0$ and $B_\rho > 0$ such that*

$$|\widehat{f}(\omega)| \leq B_\rho e^{-\rho|\omega|}, \qquad \omega \in \mathbb{R}. \tag{25}$$

**Assumption 2** (Kernel conditions K1–K4). *Fix $\varepsilon \in (0,1)$.*

    *K1. (Nonzero mass) $\widehat{K}(0) = \int_{\mathbb{R}} K(x)dx \neq 0$.*

**K2.** *(Interior-band aliasing) There exist $C_{\text{alias}}(\varepsilon) > 0$, $c_{\text{alias}}(\varepsilon) > 0$, and $p_K \geq 1$ such that for all $h > 0$ and $\gamma > 0$,*

$$\sup_{|\omega| \leq (1-\varepsilon)\pi/h} \big|1 - M_h(\omega)\big| \leq C_{\text{alias}}(\varepsilon) \exp\left(-\frac{c_{\text{alias}}(\varepsilon)}{\lambda^{p_K}}\right), \qquad M_h(\omega) := \frac{\widehat{K}_\gamma(\omega)}{D_h(\omega)}. \tag{26}$$

**K3.** *(Zero-free strip margin) There exist $\sigma > 0$ and $d_0(\sigma, \lambda) > 0$ such that the rescaled normalizer $\widetilde{D}_\lambda(\theta) := D_h(\theta/h)$ satisfies*

$$\inf_{|\Re\theta| \leq \pi,\ |\Im\theta| \leq \sigma} \big|\widetilde{D}_\lambda(\theta)\big| \geq d_0(\sigma, \lambda). \tag{27}$$

**K4.** *(Spatial localization envelope) There exist $A_K > 0$ and $a_K > 0$ such that for all $\gamma > 0$ and $x \in \mathbb{R}$,*

$$|K_\gamma(x)| \leq A_K \gamma e^{-a_K \gamma |x|}. \tag{28}$$

**Stability prefactors from K3–K4.** Assume K3 holds with parameters $(\sigma, d_0(\sigma, \lambda))$ and define

$$C_c(\lambda) := \frac{h}{d_0(\sigma, \lambda)}, \tag{29}$$

$$C_Q(\lambda) := \frac{\|K\|_{L^1(\mathbb{R})}}{h} C_c(\lambda) \coth\left(\frac{\sigma}{2}\right), \tag{30}$$

$$C_L(\lambda, \sigma) := A_K \gamma C_c(\lambda) \coth\left(\frac{a_K \lambda - \sigma}{2}\right), \tag{31}$$

where $C_L$ is defined assuming $a_K \lambda > \sigma$.

**Lemma 2** (Coefficient decay from the strip margin). *Assume K3. Then the character coefficients satisfy*

$$|c_j| \leq C_c(\lambda) e^{-\sigma|j|}, \qquad j \in \mathbb{Z}. \tag{32}$$

*Proof.* By K3, $|\widetilde{D}_\lambda(\theta)| \geq d_0(\sigma, \lambda)$ on the strip cell $\{|\Re\theta| \leq \pi,\ |\Im\theta| \leq \sigma\}$, hence $|\widetilde{C}_\lambda(\theta)| = |h/\widetilde{D}_\lambda(\theta)| \leq h/d_0(\sigma, \lambda) = C_c(\lambda)$ there. For $j \neq 0$, shift the contour in the Fourier coefficient formula for $c_j$ to $\Im\theta = \text{sign}(j)\sigma$. Then $|e^{ij\theta}| = e^{-|j|\sigma}$ and $|\widetilde{C}_\lambda(\theta)| \leq C_c(\lambda)$ on the shifted contour, yielding $|c_j| \leq C_c(\lambda) e^{-\sigma|j|}$. The case $j = 0$ is immediate. $\square$

**Lemma 3** ($L^\infty$ stability of $Q_h$). *Assume Lemma 2 and $\|K\|_{L^1(\mathbb{R})} < \infty$. Then for any $g \in L^\infty(\mathbb{R})$,*

$$\|Q_h g\|_{L^\infty(\mathbb{R})} \leq C_Q(\lambda) \|g\|_{L^\infty(\mathbb{R})}. \tag{33}$$

*Proof.* For any $x \in \mathbb{R}$,

$$|(Q_h g)(x)| \leq \|g\|_{L^\infty} \sum_{k \in \mathbb{Z}} |L_h(x - x_k)|.$$

Using $L_h(x) = \sum_j c_j K_\gamma(x - jh)$ and re-indexing,

$$\sum_{k \in \mathbb{Z}} |L_h(x - x_k)| \leq \left(\sum_{j \in \mathbb{Z}} |c_j|\right) \left(\sum_{m \in \mathbb{Z}} |K_\gamma(x - x_m)|\right).$$

The coefficient sum is bounded by Lemma 2: $\sum_j |c_j| \leq C_c(\lambda) \sum_j e^{-\sigma|j|} = C_c(\lambda) \coth(\sigma/2)$. For the kernel lattice sum, a crude bound is $\sum_{m \in \mathbb{Z}} |K_\gamma(x - x_m)| \leq \frac{1}{h} \|K_\gamma\|_{L^1(\mathbb{R})} = \frac{1}{h} \|K\|_{L^1(\mathbb{R})}$. Combining yields the stated $C_Q(\lambda)$. $\square$

**Lemma 4** (Spatial decay of the quasi-cardinal). *Assume Lemma 2 and K4. If $a_K \lambda > \sigma$, then for all $x \in \mathbb{R}$,*

$$|L_h(x)| \leq C_L(\lambda, \sigma) \exp\left(-\frac{\sigma}{h}|x|\right). \tag{34}$$

*Proof.* From $L_h(x) = \sum_j c_j K_\gamma(x-jh)$, Lemma 2, and K4,

$$|L_h(x)| \le \sum_{j\in\mathbb{Z}} C_c(\lambda)e^{-\sigma|j|} A_K \gamma e^{-a_K\gamma|x-jh|}.$$

Write $t := x/h$ so that $\gamma|x - jh| = \lambda|t - j|$ and use $|j| \ge |t| - |t - j|$ to factor out $e^{-\sigma|t|}$: $e^{-\sigma|j|}e^{-a_K\lambda|t-j|} \le e^{-\sigma|t|}e^{-(a_K\lambda-\sigma)|t-j|}$. Summing the geometric lattice tail gives $\sum_{j\in\mathbb{Z}} e^{-(a_K\lambda-\sigma)|t-j|} \le \coth((a_K\lambda-\sigma)/2)$, which yields the claim. $\square$

**Theorem 5** (Restatement of Theorem 1 with explicit constants (uniform grid))**.** *Fix $\varepsilon \in (0,1)$. Assume Assumption 1 and Assumption 2. Assume further $a_K\lambda > \sigma$ so that $C_L(\lambda,\sigma)$ is defined. Then for all $x\in\Omega$,*

$$\big|f(x) - (Q_{h,R}^{(K_c)}f)(x)\big| \le \underbrace{(1+C_Q(\lambda))\frac{B_\rho}{\pi\rho}\exp\left(-\frac{(1-\varepsilon)\pi\rho}{h}\right)}_{\text{interior resolution and filtered tail}} \tag{35}$$

$$+ \underbrace{\frac{B_\rho}{\pi\rho}C_{\text{alias}}(\varepsilon)\exp\left(-\frac{c_{\text{alias}}(\varepsilon)}{\lambda^{p_K}}\right)}_{\text{aliasing on the resolved band}} \tag{36}$$

$$+ \underbrace{\frac{2C_L(\lambda,\sigma)}{1-e^{-\sigma}}e^{-\sigma R}\|f\|_{\ell^\infty(h\mathbb{Z})}}_{\text{halo truncation}} \tag{37}$$

$$+ \underbrace{\frac{2\|K\|_{L^1(\mathbb{R})}}{h}C_c(\lambda)\frac{e^{-\sigma(K_c+1)}}{1-e^{-\sigma}}\|f\|_{\ell^\infty(\Omega_R\cap h\mathbb{Z})}}_{\text{Fourier stencil truncation (data-local)}}. \tag{38}$$

*Proof.* We use the Option A decomposition (linearity of $Q_h$):

$$f - Q_h f = (f - \mathcal{P}_{h,\varepsilon}f) + (\mathcal{P}_{h,\varepsilon}f - Q_h(\mathcal{P}_{h,\varepsilon}f)) + Q_h(f - \mathcal{P}_{h,\varepsilon}f).$$

*(i) Tail and filtered tail.* By Fourier inversion and Assumption 1,

$$\|f - \mathcal{P}_{h,\varepsilon}f\|_{L^\infty(\mathbb{R})} \le \frac{B_\rho}{\pi\rho}\exp\left(-\frac{(1-\varepsilon)\pi\rho}{h}\right).$$

Applying Lemma 3 to $g = f - \mathcal{P}_{h,\varepsilon}f$ gives the same bound multiplied by $C_Q(\lambda)$, yielding the first line equation 35. *(ii) Aliasing.* On the band $|\omega| \le (1-\varepsilon)\pi/h$, the operator has Fourier multiplier $M_h(\omega)$, so $\widehat{\mathcal{P}_{h,\varepsilon}f - Q_h(\mathcal{P}_{h,\varepsilon}f)}(\omega) = (1 - M_h(\omega))\widehat{f}(\omega)$ on that band. Using Assumption 1 and K2 gives equation 36. *(iii) Truncations.* For halo truncation, write $(Q_h - Q_{h,R})f$ as a sum over $k \notin \{-R,...,N+R\}$ and apply Lemma 4, bounding the two geometric tails by $2e^{-\sigma R}/(1 - e^{-\sigma})$, yielding the halo term in equation 38. For stencil truncation, write $L_h - L_h^{(K_c)} = \sum_{|j|>K_c} c_j K_\gamma(\cdot - jh)$, use Lemma 2 to bound $\sum_{|j|>K_c}|c_j| \le 2C_c(\lambda)e^{-\sigma(K_c+1)}/(1-e^{-\sigma})$, and use $\|K_\gamma\|_{L^1} = \|K\|_{L^1}$ to obtain the last term in equation 38. Combining (i)–(iii) yields the result. $\square$

A.1.3   KERNEL VERIFICATION FOR THE NORMALIZED sech$^2$ FAMILY

We verify Assumption 2 for the normalized profile

$$K(x) := \frac{1}{2}\text{sech}^2(x), \qquad K_\gamma(x) = \frac{\gamma}{2}\text{sech}^2(\gamma x). \tag{39}$$

**K1 (mass) and K4 (envelope).** Since $\int_{\mathbb{R}} \text{sech}^2(x)\, dx = 2$, we have $\widehat{K}(0) = \int K = 1$. Also $\text{sech}(t) \le 2e^{-|t|}$ implies

$$|K_\gamma(x)| = \frac{\gamma}{2}\text{sech}^2(\gamma x) \le 2\gamma e^{-2\gamma|x|}, \tag{40}$$

so K4 holds with $(A_K, a_K) = (2,2)$.

**Closed-form Fourier transform.**

**Proposition 1** (Fourier transform of $\frac{1}{2}\mathrm{sech}^2$)**.** *With the convention in Appendix A.1.1,*

$$\widehat{K}(\xi) = \frac{\frac{\pi}{2}\xi}{\sinh\left(\frac{\pi}{2}\xi\right)}, \qquad \xi \in \mathbb{R}, \tag{41}$$

*with the continuous extension $\widehat{K}(0) = 1$. Consequently $\widehat{K}(\xi) > 0$ for all $\xi \in \mathbb{R}$.*

*Proof.* It suffices to compute $I(\xi) := \int_{\mathbb{R}} \mathrm{sech}^2(x) e^{-i\xi x}\, dx$ and then divide by 2. Fix $\xi > 0$ and consider $f(z) := \frac{e^{-i\xi z}}{\cosh^2 z}$ integrated over the rectangle with vertices $\pm R$ and $\pm R + i\pi$. Using $\cosh(z + i\pi) = -\cosh(z)$, we have $f(z + i\pi) = e^{\xi\pi} f(z)$. Letting $R \to \infty$ (vertical sides vanish by exponential decay), residue calculus gives

$$(1 - e^{\xi\pi}) I(\xi) = 2\pi i \mathrm{Res}\big(f; z_0 = i\pi/2\big),$$

where $z_0 = i\pi/2$ is the only pole in the strip. Since $\cosh z \sim i(z - z_0)$ near $z_0$, we have $f(z) \sim -e^{-i\xi z}(z - z_0)^{-2}$, a second-order pole with residue $\mathrm{Res} = \frac{d}{dz}\big(-e^{-i\xi z}\big)\big|_{z=z_0} = i\xi e^{\xi\pi/2}$. Thus $(1 - e^{\xi\pi}) I(\xi) = -2\pi\xi e^{\xi\pi/2}$, i.e. $I(\xi) = \pi\xi/\sinh(\pi\xi/2)$. Evenness gives the same formula for $\xi < 0$, and dividing by 2 yields the stated $\widehat{K}$. $\qquad\square$

**K2 (interior-band aliasing) with $p_K = 1$.**

**Proposition 2** (Interior-band aliasing for $\mathrm{sech}^2$)**.** *Fix $\varepsilon \in (0,1)$. There exists $C_{\mathrm{alias}}(\varepsilon) > 0$ such that for all $h > 0$, $\gamma > 0$,*

$$\sup_{|\omega| \leq (1-\varepsilon)\pi/h} \big|1 - M_h(\omega)\big| \leq C_{\mathrm{alias}}(\varepsilon) \exp\left(-\frac{\varepsilon\pi^2}{\lambda}\right), \qquad \lambda = \gamma h. \tag{42}$$

*In particular, K2 holds with $p_K = 1$ and one may take $c_{\mathrm{alias}}(\varepsilon) = \varepsilon\pi^2$.*

*Proof.* By Proposition 1, $\widehat{K}_\gamma(\omega) \geq 0$, hence $D_h(\omega) \geq \widehat{K}_\gamma(\omega)$ and

$$|1 - M_h(\omega)| = \frac{\sum_{m \neq 0} \widehat{K}_\gamma(\omega + 2\pi m/h)}{D_h(\omega)} \leq \sum_{m \neq 0} \frac{\widehat{K}_\gamma(\omega + 2\pi m/h)}{\widehat{K}_\gamma(\omega)}.$$

Write $\widehat{K}_\gamma(\omega) = \widehat{K}(\omega/\gamma)$ and set $F(t) := \widehat{K}(t)$ for $t \geq 0$. On the interior band, $|\omega| \leq (1 - \varepsilon)\pi/h$ implies $t := |\omega|/\gamma \leq (1 - \varepsilon)\pi/\lambda$. For $m \geq 1$, the closest approach is at $\omega = -(1 - \varepsilon)\pi/h$, giving $|\omega + 2\pi m/h|/\gamma \geq (2m - 1 + \varepsilon)\pi/\lambda$, hence the gap $\delta_m \geq 2(m - 1 + \varepsilon)\pi/\lambda$. Using $F(t) = \frac{(\pi/2)t}{\sinh((\pi/2)t)}$ and the inequality $\sinh(a + b) \geq \frac{1}{2} e^b \sinh(a)$ for $a, b \geq 0$, one obtains $F(t + \delta)/F(t) \leq C(\varepsilon) e^{-(\pi/2)\delta}$ uniformly for $t \in [0, (1 - \varepsilon)\pi/\lambda]$ (after absorbing the mild $(t + \delta)/t$ factor into $C(\varepsilon)$). Thus

$$\frac{\widehat{K}_\gamma(\omega + 2\pi m/h)}{\widehat{K}_\gamma(\omega)} \leq C(\varepsilon) \exp\left(-\frac{\pi}{2}\delta_m\right) \leq C(\varepsilon) \exp\left(-\frac{(m - 1 + \varepsilon)\pi^2}{\lambda}\right).$$

Summing the geometric series over $m \geq 1$ and using symmetry for $m \leq -1$ yields the claimed bound. $\quad\square$

**K3 (zero-free strip margin) via diagonal dominance.** We include a convenient sufficient condition that makes the $\lambda$–$\sigma$ feasibility explicit.

**Proposition 3** (Poisson form of the normalizer)**.** *Assume $K_\gamma$ decays sufficiently fast for Poisson summation. Then*

$$D_h(\omega) = h \sum_{k \in \mathbb{Z}} K_\gamma(kh) e^{-i\omega kh}, \qquad \widetilde{D}_\lambda(\theta) = D_h(\theta/h) = \lambda \sum_{k \in \mathbb{Z}} K(\lambda k) e^{-ik\theta}. \tag{43}$$

*Proof.* This is the standard Poisson summation identity applied to $x \mapsto K_\gamma(x) e^{-i\omega x}$. $\qquad\square$

**Target Functions for Approximation**
**(Shaded region = training domain [-1, 1])**

Figure 8: Target functions evaluated in Fig 5c

**Proposition 4** (Diagonal dominance sufficient condition for K3: $\mathrm{sech}^2$). *Let* $K(x) = \frac{1}{2}\mathrm{sech}^2(x)$. *Fix* $\sigma > 0$ *and assume* $2\lambda > \sigma$. *Then for all* $\theta \in \mathbb{C}$ *with* $|\Im\theta| \leq \sigma$,

$$\left|\widetilde{D}_\lambda(\theta)\right| \geq \frac{\lambda}{2}\Big(1 - q(\sigma,\lambda)\Big), \qquad q(\sigma,\lambda) := \frac{8e^{-(2\lambda-\sigma)}}{1 - e^{-(2\lambda-\sigma)}}. \tag{44}$$

*In particular, if* $q(\sigma,\lambda) < 1$ *then K3 holds with* $d_0(\sigma,\lambda) \geq \frac{\lambda}{2}(1 - q(\sigma,\lambda)) > 0$.

*Proof.* By Proposition 3, $\widetilde{D}_\lambda(\theta) = \lambda\sum_{k\in\mathbb{Z}} K(\lambda k)e^{-ik\theta}$. Write $\theta = \theta_r + i\theta_i$ with $|\theta_i| \leq \sigma$. Then $|e^{-ik\theta}| = e^{k\theta_i} \leq e^{\sigma|k|}$. Separating the $k=0$ term and applying the triangle inequality,

$$|\widetilde{D}_\lambda(\theta)| \geq \lambda K(0) - \lambda\sum_{k\neq 0}|K(\lambda k)|e^{\sigma|k|}.$$

Using $|K(\lambda k)| \leq 2e^{-2\lambda|k|}$ gives $|K(\lambda k)|e^{\sigma|k|} \leq 2e^{-(2\lambda-\sigma)|k|}$, so $\sum_{k\neq 0}|K(\lambda k)|e^{\sigma|k|} \leq 4\sum_{k\geq 1}e^{-(2\lambda-\sigma)k} = 4\frac{e^{-(2\lambda-\sigma)}}{1-e^{-(2\lambda-\sigma)}}$. Since $K(0) = 1/2$, the claim follows. $\square$

A.2   TARGET FUNCTIONS

