# OpenReview forum: "Constructing Machine-Precision Neural Networks with Quasi-Interpolants"
_ICLR.cc/2026/Workshop/FM4Science — ICLR 2026 Workshop FM4Science Poster_

### Official Review · Reviewer_vHwQ · 2026-02-13
**Constructing Machine-Precision Neural Networks with Quasi-Interpolants - Clear Accept**

**Rating:** 8
**Confidence:** 4

**Review:**

The paper addresses the "precision ceiling" in scientific machine learning, where neural networks fail to reach machine precision (e.g., $10^{-16}$ for fp64) even on simple, noise-free interpolation tasks. The authors propose a novel construction for one-hidden-layer MLPs based on quasi-interpolation theory. They prove that this construction achieves machine precision with $O(\log(1/\epsilon))$ parameter scaling while remaining numerically stable in floating-point arithmetic. By comparing their construction to end-to-end trained models, they demonstrate that the performance gap is due to optimization failures - specifically, the failure of standard training to discover the required weight-scaling and bandwidth properties.

Strengths

S1. Theoretical Contribution: The paper provides the first explicit MLP construction that matches the optimal $O(\log(1/\epsilon))$ parameter scaling of classical polynomial methods while remaining implementable in floating-point arithmetic. It successfully bridges the gap between universal approximation theory (which often uses non-realizable weights) and practical numerical analysis.

S2. Diagnostic Utility: The construction identifies a critical dimensionless bandwidth parameter, $\lambda = \gamma h$, which reveals why standard training fails: trained networks drive $\lambda \to 0$ as width increases, whereas the high-precision construction requires $\lambda$ to remain roughly constant.

S3. Rigorous Empirical Validation: The authors demonstrate that when the "QI geometry" is fixed (grid and bandwidth), MLPs can indeed reach the fp64 floor ($10^{-15}$ to $10^{-16}$ relative error) via simple coefficient fitting, proving that expressivity is not the bottleneck.

S4. Insightful Analysis of Training Pathologies: The paper uses pruning (Orthogonal Matching Pursuit) and Hessian analysis to show that trained networks learn redundant, sparse representations and exhibit "rank saturation," even though the more precise QI-constructed solutions are actually better-conditioned.

Weaknesses

W1. Limited Dimensionality: The analysis and construction are primarily focused on 1D interpolation tasks. While this is a standard starting point for numerical analysis, the extension to high-dimensional scientific problems (typical of foundation models and thus of relevance for this workshop) is not explicitly developed or tested.

W2. Practical Optimization Gap: While the paper identifies why optimization fails (e.g., weight scaling mismatch), it does not provide a new optimizer or training recipe that allows standard MLPs to discover these high-precision solutions from scratch without the explicit QI parameterization.

W3. Assumptions on Target Smoothness: The geometric convergence rates ($e^{-c/h}$) rely on the target function being analytic. The paper does not thoroughly explore how performance degrades for non-analytic or less smooth functions (e.g., $C^k$ functions) often found in real-world physics.

W4. Grid Uniformity: The construction relies on uniform samples on a grid. In many scientific ML applications, data is irregularly sampled or resides on unstructured meshes, and it is unclear how the Fourier-normalization approach would adapt to these cases.

I recommend accepting this paper (8). It provides a significant theoretical advance by identifying a numerically realizable path to machine precision in MLPs. It offers deep insights into the "optimization vs. expressivity" debate in scientific machine learning, making it a strong fit for a workshop focused on the foundations of AI for science.

---

### Official Review · Reviewer_CqaA · 2026-02-16

**Rating:** 4
**Confidence:** 2

**Review:**

**Summary:**

This paper studies whether MLPs can achieve machine-precision accuracy when interpolating smooth 1D functions on a fixed grid. The authors provided theoretical guarantees on approximation error and analyzed why standard training often fails to reach machine precision even when the network is expressive enough. Experiments are conducted on synthetic 1D functions sampled uniformly on [-1, 1], demonstrating that the constructed networks can achieve near machine-level interpolation accuracy. The focus of the paper is theoretical and numerical, but it fails to connect the proposed framework to either foundation models or specific scientific studies.


**Strengths:**

1. This paper shows how quasi-interpolants can be implemented within shallow MLPs and derives parameter scaling results for achieving a small approximation error. The presentation of the approximation result is technically careful.

2. The authors follow a clear, rigorous and complete procedure to formulate research questions about machine-precision expressivity of MLPs and optimization solutions, construct the framework, provide mathematical analyses, and design controlled numerical experiments.

3. The discussion on why gradient-based training does not recover machine-precision solutions is interesting. This may be relevant for numerical analysis and scientific computing communities concerned with high-precision computation.


**Weaknesses:**

1. While the technical contribution is interesting from an approximation-theoretic perspective, the paper does not clearly align with the workshop’s scope on scientific foundation models. It does not demonstrate how machine-precision interpolation would impact real scientific workflows, nor does it apply the suggested framework to scientific scenarios to address specific scientific domain problems.

2. The experiments are limited to noiseless 1D interpolation of smooth functions, using a small number of uniformly spaced samples and simple MLP models. Results thereby remain in a synthetic numerical regime rather than being validated in a domain-specific scientific task. It is unclear whether the quasi-interpolant construction scales to high-dimensional settings, remains valid under typically noisy scientific data, or applies to large-scale foundation models.

---

### Meta-Review · Area_Chair_wbzo · 2026-02-27

**Recommendation:** Accept (Poster)
**Confidence:** 4

**Metareview:**

The average review score is above 6, which means reviewers recommended an acceptance.

---

### Decision · Program_Chairs · 2026-03-03

Accept (Poster)